# Genetic Diversity and Structure for Conservation Genetics of Goldeye Rockfish *Sebastes thompsoni* (Jordan and Hubbs, 1925) in South Korea

**DOI:** 10.3390/biology14111559

**Published:** 2025-11-06

**Authors:** Kang-Rae Kim, Keun-Sik Kim, Sung Jin Yoon

**Affiliations:** 1Southeast Sea Fisheries Research Institute, National Institute of Fisheries Science, Namhae 52440, Republic of Korea; kimkangrae9586@gmail.com; 2Restoration Research Team (Fishes/Amphibians & Reptile), Research Center for Endangered Species, National Institute of Ecology, Yeongyang 36531, Republic of Korea; kskim@nie.re.kr; 3Ulleungdo-Dokdo Ocean Science Station, Korea Institute of Ocean Science & Technology, Ulleung 40205, Republic of Korea

**Keywords:** microsatellite, bottleneck, population structure, genetic diversity, migration rate

## Abstract

This study assessed the genetic status of the rockfish *Sebastes thompsoni* from five coastal sites along South Korea and East Seas sampled in 2018. Using microsatellite markers, we measured genetic diversity and differences among locations. We found consistently high diversity and minimal genetic separation among regions, indicating one large, connected population. However, effective population size was mostly below 1000, suggesting a risk of future diversity loss if pressures persist. These results support managing *S. thompsoni* as a single unit across the South and East Seas, while prioritizing habitat protection, controls on overfishing, use of diverse broodstock in any release programs, and regular genetic monitoring. Our findings provide practical evidence to guide the sustainable use of this ecologically and economically important species in Korean waters.

## 1. Introduction

*Sebastes thompsoni* Jordan and Hubbs, 1925 (goldeye rockfish), a cold-water reef fish, is widely distributed along the northwestern Pacific coast, inhabiting complex reef and bay habitats from the East Sea to the South Sea of the Korean Peninsula [1]. Goldeye rockfish is valuable for commercial and artisanal fisheries in Korea and is mainly consumed along the southern coast. Despite its broad distribution, little is known about how historical environmental changes have affected *S. thompsoni* populations in Korean waters. Historic environmental changes, such as past climate shifts or human fishing pressures, can diminish genetic diversity and alter the effective population size, thereby impacting the long-term viability of a species [2,3]. Climatic fluctuations modulate sea temperature and current regimes that govern pelagic larval transport and settlement along the Korean coasts [4,5]. Such changes can drive cohort failure or expansion, habitat contraction of demersal adults, and connectivity breakdowns, leading to bottlenecks and temporal declines in effective population size [4,5]. Consequently, estimating the historically effective population size of the goldeye rockfish through genetic analysis provides crucial scientific evidence for understanding how they have responded to past environmental changes and for assessing the current genetic health of their population [6,7]. A previous microsatellite-based survey reported moderate heterozygosity (mean *H*_O_ = 0.615) and weak population structure in samples collected between 2011 and 2014 [4,8]. However, these studies focused on genetic differentiation and lacked essential indicators for genetic management, such as modern effective population size (*N*_e_), formal marker quality control for null alleles and stutters, and reconstruction of past *N*_e_ [4]. Furthermore, understanding of the causes of population fluctuations, particularly the bottlenecks evident in historical reconstructions, remains inadequate. Past climate variability alters sea temperature and the strength of boundary currents, which can reduce *N*_e_ and trigger bottlenecks; in turn, loss of genetic diversity and low *N*_e_ may weaken long-term viability [9,10,11]. Therefore, identifying historical bottlenecks helps time these events and prepare for climate-driven impacts [12].

*N*_e_ is a key indicator of maintaining genetic diversity in a population and is therefore essential for establishing species conservation strategies [12,13]. Reconstructing past *N*_e_ fluctuations can help infer the impact of environmental factors such as climate change on populations, while current *N*_e_ provides important baseline data for assessing population health and establishing practical management plans [13,14,15].

Genetic structure is an essential element in establishing conservation units [16]. Typically, MUs are established when significant genetic and ecological differentiation is demonstrated. If differentiation is not confirmed, it is appropriate to treat the range as a single management unit [16,17,18,19]. This principle also applies to fish resource management [20]. Prior studies, while not explicitly defining MUs, have suggested an indirect division into two populations [4]. Based on recent data, the actual differentiation needs to be re-evaluated to propose appropriate management units for *S. thompsoni*.

Methods for analyzing genetic diversity include microsatellite DNA, mitochondrial DNA, and single-nucleotide polymorphisms (SNPs) [21,22]. Among these, microsatellite DNA remains one of the most widely used tools in population genetic studies because it provides a relatively simple and cost-effective way to assess genetic diversity and population structure [23]. Microsatellites are multi-allelic and mutate faster than SNPs, so a small, well-chosen panel delivers higher polymorphic information content and greater power to detect weak structure, recent bottlenecks, and fine-scale relatedness than an equivalently sized SNP set [24]. In fish, microsatellites capture variation among individuals through differences in short repetitive sequences, and their high level of polymorphism makes them particularly valuable for such analyses [25,26,27].

In this study, we used seven microsatellites were employed to (1) quantify current genetic diversity and inbreeding (*H*_O_, *H*_e_, *F*_IS_) across five Korean populations sampled simultaneously in 2018; (2) analyze population structure via STRUCTURE and DAPC analyses; and (3) Assess genetic flow between populations and reconstruct historical effective population sizes within 10,000 generations using VarEff. We aim to evaluate the effective population size and genetic diversity of the current population through genotypic data and provide basic data for management as aquatic resources.

## 2. Materials and Methods

### 2.1. Sampling and DNA Extraction

*S. thompsoni* specimens were collected during May 2018 at five key locations selected to cover the genetic breadth, primarily using bottom gillnets and fish traps, at water depths of 70–150 m (Figure 1, Appendix A). Whole fish were preserved immediately in 99.9% ethanol. For DNA extraction, 10 mg of caudal fin tissue was excised and submerged in fresh 99.9% ethanol. Following Asahida et al. [28] tissues were incubated at 55 °C for 12 h in TNES urea buffer (8 M urea; 10 mM Tris HCl, pH 7.5; 125 mM NaCl; 10 mM EDTA; 1% SDS) supplemented with proteinase K (100 μg/mL, Sigma, St. Louis, MO, USA). After digestion, proteins were removed by phenol:chloroform:isoamyl alcohol (25:24:1) extraction, and nucleic acids were precipitated with 2-propanol. Pellets were washed in 70% ethanol, air-dried, and dissolved in sterile triple-distilled water. DNA concentration was adjusted to 50 ng/μL, and samples were stored at −20 °C until microsatellite PCR.

### 2.2. Microsatellite Genotyping

The microsatellite markers used in this study are Sth3A, Sth24, Sth45, KSs6, KSs2A, Sth91, and Sth37 [8,29]. Details are provided in Appendix A. Seven markers were tested for stutter using Micro-Checker, and no evidence of a stutter was observed. In addition, although no evidence of null allele was observed in Micro-Checker [30], the null frequency was less than 0.05 when verified using FreeNA (Sth3A, Sth24, KSs6) [31], and did not exceed 0.05, and the comparison results of *F*_ST_ and *F*_ST_ ENA showed very minor differences, showing that the seven markers are suitable for population genetic analysis (Appendix A).

PCR was performed on a Mastercycler^®^ pro (Hamburg, Germany, Eppendorf) using 10 ng of genomic DNA in AccuPower^®^ PCR PreMix (Bioneer, Daejeon, Republic of Korea) with 0.8 μM forward primer (FAM, VIC, NED, or PET labeled) and 0.8 μM reverse primer. PCR condition consisted of an initial denaturation at 94 °C for 5 min; 34 cycles of 94 °C for 30 s, 58 °C for 45 s, and 72 °C for 30 s; followed by a final extension at 72 °C for 15 min and a hold at 4 °C. Amplicons were verified on 1.5% agarose gels to confirm band presence and fragment size. For genotyping, PCR products were mixed with GeneScan™ 500 ROX size standard and Hi-Di™ formamide (Applied Biosystems, Foster City, CA, USA), denatured at 95 °C for 2 min, and immediately cooled to 4 °C. Allele sizes were determined on an ABI 3730xl DNA Analyzer (Applied Biosystems), and genotypes were called using GeneMarker^®^ (ver. 2.6.7, SoftGenetics, State College, PA, USA).

### 2.3. Genetic Diversity Analyses by Microsatellite Markers

We assessed potential scoring errors at microsatellite loci with MICROCHECKER (ver. 2.2.3) [30]. Genetic diversity was assessed as the number of alleles (*N*_A_), expected heterozygosity (*H*_E_), and observed heterozygosity (*H*_O_) using CERVUS (ver. 3.0) [32]. Analyses of the population inbreeding coefficient (*F*_IS_) and Hardy–Weinberg equilibrium (HWE) were conducted with GENEPOP (ver. 4.2) [33] and ARLEQUIN (ver. 3.5) [34]. Bottleneck signatures were evaluated with two complementary approaches. First, we used BOTTLENECK (ver. 1.2.02) [35] to test for heterozygosity excess, and second, we considered the infinite alleles model (IAM) [36]. For these estimations, a two-phase model (TPM) and a stepwise mutation model (SMM) [37] were applied, with the TPM configured to 10% variance and 90% SMM. Microsatellite evolution largely follows a generalized stepwise model in which single-step slippage predominates, with occasional multistep changes [38,39]. Each model was run for 10,000 iterations, and significance was assessed using the Wilcoxon signed-rank test [40]. Effective population size (*N*_e_) for each population was estimated in NeEstimator (ver. 2.1) [41] the linkage-disequilibrium method (LDNe) under a random-mating model [42]. The minimum allele frequency (MAF) threshold used in LDNe calculations is 0.02.

### 2.4. Population Genetic Structure Analysis and Migration Rate

Genetic differentiation among groups and molecular variance (AMOVA) were evaluated in ARLEQUIN (ver. 3.5) [34]. AMOVA used 1000 permutations. Bayesian clustering of genetic structure was performed in STRUCTURE (ver. 2.3.4) [43]. We explored *K* values from 1 to 10 under an admixture model appropriate for mixed water systems to identify the best-supported grouping. For each *K*, 10 independent runs were executed with a burn-in of 10,000 iterations followed by 100,000 MCMC iterations. The most likely *K* was inferred via the Δ*K* approach of Evanno et al. [43] using STRUCTURE SELECTOR (https://lmme.ac.cn/StructureSelector/ accessed 10 April 2025). Discriminant analysis of principal components (DAPC) was performed in R with the ADEGENET package (ver. 2.1.3) [44]. When performing DAPC, the genetic differentiation signal was very weak, resulting in an ambiguous BIC curve for “find.clusters”. Because BIC is sensitive to the number of PCs retained and the k-means assumption, we did not report BIC-based *K* selection to avoid over segmentation due to reliance on an ambiguous BIC minimum. Instead, we visualized the PCs using a predefined sample population (pop) set to 40.

Bayesian gene flow estimation was performed for 5 populations and 7 microsatellite loci (153 individuals in total) using MIGRATE-n (ver. 4.4.5) [45]. The mutation process was assumed to be a Brownian stepwise mutation model, and the prior distribution for each parameter was set to Θ (4 Nμ) with a uniform distribution of 0–1000 and the migration rate M = m/μ with a uniform distribution of 0–100. MCMC was run with four long chains, and the genetic tree was saved every 100 steps while additionally executing 500,000 steps after the burn-in of 1,000,000 steps (5000 samples per chain, 20,000 posterior samples in total). The acceptance rates by chain were in the recommended range of 0.32–0.66 (mean 0.47), and convergence was confirmed with an effective sample size (ESS) ≥ 600 for all Θ M parameters (range 602–1720; 580 ± 20). In addition, visual inspection of the trace plot and the Gelman Rubin statistic (R^ < 1.05) supported the convergence. The number of migrants per generation (Nm) was converted and interpreted using the formula Nm = M × Θ⁄4 assuming μ = 10^−3^. The final migration rates are presented in Appendix A.

### 2.5. Historical Effective Population Size Analysis

We inferred historical demographic trajectories and posterior distributions of key parameters by combining coalescent theory with Markov chain Monte Carlo sampling, an approach shown to be robust under realistic mutation-model assumptions and moderate bottleneck violations [46,47]. Specifically, we used the VarEff (ver. 1.2) R package [15] to model changes in effective population size (*N*_e_) from the present back to 2000 generations ago based on nuclear microsatellite loci. VarEff (ver. 1.2) R package [15] approximates the data likelihood under a stepwise mutation model (SMM) and employs MCMC to sample piecewise-constant *N*_e_ trajectories, thereby reconstructing the posterior distribution of the time to most recent common ancestor (TMRCA) and identifying periods of demographic contraction or expansion. Peaks in the resulting posterior distributions mark probable coalescence times, while plateaus or troughs indicate the duration of bottleneck events. We set the per-locus mutation rate (μ) to 5 × 10^−4^ typical for marine species and assumed a generation time (G) of six years for *S. thompsoni* [8,48]. We set the per-locus mutation rate to μ = 5 × 10^−4^, a mid-range value commonly reported for fish microsatellites and assumed a generation time of G = 6 years for *S. thompsoni* based on published age–growth information [48].

## 3. Results

### 3.1. Microsatellite Genetic Diversity

Seven microsatellite loci and their allele frequencies were analyzed for genetic diversity indices across five populations (Table 1). The average number of alleles, allelic richness, observed heterozygosity (*H*_O_), and expected heterozygosity (*H*_E_) ranged from 6.3 to 7.3, 6.29 to 6.55, 0.759 to 0.816, and 0.659 to 0.699, respectively. Five populations deviated from HWE. In all populations, the inbreeding index was negative, and *F*_IS_ was significant in the YD population (*p* < 0.05). The observed heterozygosity was highest in the YD population (*H*_O_ = 0.816) and lowest in the SA population (*H*_O_ = 0.759).

### 3.2. Bottleneck Analysis

Using infinite allele model (IAM), we identified significant bottlenecks in all populations (*p* < 0.05). Two-phase model (TPM) identified bottlenecks in all populations (Table 2). All populations showed recent mode shifts, indicating evidence of bottlenecks. The effective population sizes of the five populations ranged from 108 to 254. The YD population had the smallest effective population size of 108 (Table 2). Except for the TY population (not estimated), remaining populations had effective population sizes of less than 1000.

### 3.3. Population Structure, Genetic Differentiation Analyses and Gene Flow

In the microsatellite dataset, most *F*_ST_ values were not statistically significant, and all *F*_ST_ values between populations were less than 0.01, indicating very low genetic differentiation (Table 3). STRUCTURE analysis maximized the delta *K* value for the population structure at *K* = 7 and 9 (Figure 2). According to the STRUCTURE results, *K* = 1, which has an L(K) value close to 0, is the most suitable model (Table 4). The results of DAPC, which analyzes the population structure based on a model-less method, showed that each population (BS, SA, TY, UL, YD) was mixed and formed into one population, unlike the results of STRUCTURE analysis (Figure 2 and Figure 3). The STRUCTURE software showed panmixia at both *K* = 7 and 9, indicating that all populations belong to a single, genetically homogeneous group (Figure 3).

To investigate the genetic structure of *S. thompsoni*, AMOVA was performed on five populations (Table 5). AMOVA was 0.13% for Among groups and 99.87% for Within populations. AMOVA showed 99.87% of within-population variance, suggesting that it is a single population.

MIGRATE-n analysis revealed that gene flow between populations was generally low, but there was a clear directionality (Appendix A). The most prominent value in the posterior mean migration rate (M = m/μ) matrix was YD → TY (M = 19.11), which was the largest among all paths and more than three times higher than the reverse direction (TY → YD; M = 5.75). In addition, UL consistently showed strong inflows to all other populations, with M values of UL → SA, UL → BS, and UL → YD of 15.31, 14.94, and 15.97, respectively, falling within the top 10% flow range (Appendix A). In contrast, outflows from BS, SA, and TY were relatively weak, with M ≤ 11 for most of them.

### 3.4. Analysis of Historical Effective Population Size

VarEff-based generational *N*_e_ analysis revealed that the five populations of *S. thompsoni* expanded maximally in generations 100–200 within 10,000 generations and then declined in recent generations (Figure 4 and Appendix A). Although there were some generational differences among the populations, the decline in *N*_e_ was generally recent.

## 4. Discussion

### 4.1. Genetic Diversity and Population Bottleneck

In our study, the genetic diversity of *S. thompsoni* was high (*H*_O_ = 0.759–0.816), which is higher than the average *H*_O_ = 0.615 and 0.709 reported for *S. thompsoni* and *S. schlegelii*, respectively, in closely related species [4,38]. This discrepancy likely reflects differences in marker polymorphism, specifically the number of alleles scored per locus, rather than true biological divergence. Although our sampling sites differed somewhat from those of the previous study, all specimens in our work were collected simultaneously in 2018, reducing temporal bias. Moreover, DeFaveri and Merilä [49] found no significant effect of sampling period on MS-based diversity estimates. Unlike previous studies, this study excluded four of the eleven markers used in the previous study due to null and stutter data. These null and stutter data could potentially lower the *H*_O_ [4]. Therefore, it is believed that the differences in *H*_O_ are due to the characteristics of these markers [50,51]. The negative *F*_IS_ pattern observed in most markers is an artificial signal indicating strong recent gene flow, and these *H*_O_ patterns are considered biological indicators.

Despite difficulties in directly comparing absolute *H*_O_ values across studies, the negative *F*_IS_ values we observed suggest a higher level of gene flow among these populations compared to earlier sampling periods. Negative *F*_IS_ indicates an excess of heterozygotes consistent with immigration from external sources [52]. Thus, the 2018 population appears to experience stronger populations connectivity than those sampled between 2011 and 2014. In population genetics, when the estimated number of migrants per generation (*N*_e_*m*) exceeds 1, there is considered to be sufficient genetic flow to offset population differentiation due to genetic drift [53,54]. This high genetic flow in marine fish is often due to extensive larval dispersal [55].

Effective population size (*N*_e_) buffers genetic diversity against loss over generations [11] and is generally recommended to exceed 1000 to maintain long-term evolutionary potential [12]. Historical *N*_e_ reconstructions also revealed a significant bottleneck, corroborated by the sharp recent decline in *N*_e_. From a fisheries management perspective, reduced *N*_e_ undermines the sustainability of harvests and heightens extinction risk. In this study, a high *H*_O_ merely reflects current admixture and marker polymorphism, not safety from genetic drift. Since *N*_e_ < 1000 in all five populations, heterozygosity declines at a rate of approximately 1/(2 *N*_e_) per generation [11]. For *N*_e_ of 100–250, a decline of approximately 0.2–0.5% per generation is expected [11,12]. The recent bottleneck suggested by the VarEff trajectory suggests that the current high *H*_O_ may be a temporary indicator, and allelic richness and adaptive potential are likely already in a declining phase. Therefore, our results underscore the need for conservation actions aimed at increasing *N*_e_ through measures such as habitat protection, reducing overexploitation, and, where appropriate, facilitating translocations to bolster genetic diversity in both southern and eastern sea populations of *S. thompsoni*.

### 4.2. Population Genetic Structure

The five *S. thompsoni* populations examined in this study exhibited minimal genetic differentiation: STRUCTURE analysis revealed panmixia, and DAPC clearly clustered all samples into a single group. When the genetic structure signal is weak or the population is actually a single entity, STRUCTURE can induce over segmentation due to unstable *K* estimation, and cluster estimation can be biased, especially when the sample size is imbalanced [43,56]. Since Δ*K* is an indicator based on the second difference of lnP(D|K), it is calculated only when *K* = 2 or higher, and by design, it cannot evaluate *K* = 1 [26,43,56]. Therefore, the presence of a single structure should be judged not by Δ*K* but by the trend of lnP(D|K) and the homogeneity of the bar plot [26,57,58]. In such situations, Δ*K* tends to emphasize only the upper level of *K* > 1, which can be misleading. Therefore, cross-validation with an independent indicator such as DAPC, AMOVA, or *F*_ST_ is recommended [26,57,58]. This pattern is further supported by the negative *F*_IS_ values, which point to ongoing gene flow from external sources that homogenizes genetic variation among populations. Although previous studies have reported differences between Dokdo and other regions, this study did not include Dokdo due to limitations in population collection, and no significant differences were found among the five current populations [4].

Such connectivity likely arises because there are no major geographic or ecological barriers between the southern and eastern seas of the Korean Peninsula; currents and larval dispersal promote continual exchange of individuals [4]. In the absence of barriers to gene flow, localized genetic divergence cannot establish, and the resulting genetic homogeneity can be both a blessing and a threat. On one hand, panmixia simplifies conservation and restoration efforts: any individual can be translocated among regions with minimal risk of maladaptation to local genotypes [59]. This flexibility facilitates the design of broad, integrated management plans rather than requiring region-specific breeding and release programs.

On the other hand, a single, well-mixed population offers no “backup” if it suffers a large-scale disturbance. Species that maintain multiple, semi-independent populations may endure the loss of one sub-population without endangering the species as a whole, but panmictic stocks like *S. thompsoni* lack such redundancy [60]. A sudden habitat degradation, or overexploitation could therefore impact the entire genetic reservoir simultaneously. To mitigate these risks, ongoing genetic monitoring and habitat protection throughout the species range remain essential, even as we take advantage of the benefits conferred by its genetic unity.

In addition to the lack of major geographical barriers, the two-phase life history of *Sebastes* appears to support the observed connectivity [5]. Pelagic larvae are dispersed along local circulation and currents, after which adults are mostly benthic and prefer to settle in rock habitats [4]. These initial dispersal stages and the more characteristic habitat preferences of adults may provide high genetic connectivity, as seen in the microsatellite markers [4,5]. Management must, therefore, (i) protect the nursery and settlement habitats utilized by pelagic larvae, (ii) establish protected or management areas spaced within the typical larval dispersal range expected from local currents, and (iii) identify adult movement ranges and maintain habitat continuity. These characteristics, along with the STRUCTURE and DAPC results, are consistent with panmixia across the sampled scale but still necessitate spatially explicit conservation measures that secure both larval supply and adult habitat quality.

### 4.3. Historical Effective Population Size

The cold-water rocky reef fish *S. thompsoni* is widely distributed along the coastal waters of the northwestern Pacific [1]. Population genetic analyses in this study suggest that this species experienced the *N*_e_ expansion approximately 600–1200 years ago, assuming a generation time of six years (100–200 generations ago) [61]. This period corresponds to the climatic transition from the Medieval Warm Period to the Little Ice Age, when a general trend of cooling began [62,63]. Although *S. thompsoni* is regarded as a cold-adapted rock fish, the inferred *N*_e_ expansion 600–1200 years ago is better framed as counterintuitive rather than contradictory [1]. During the late Medieval Warm Period, a temporary strengthening of the northward-flowing Tsushima Current likely intensified the East Korea Warm Current, elevating surface and subsurface temperatures in the Southeast and East Sea relative to adjacent waters [9,10,64,65,66]. Such hydrographic changes could have created thermally optimal habitats for growth and spawning while simultaneously boosting marine primary productivity and the abundance of planktonic prey, thereby enhancing recruitment and driving a net population increase [65]. In other words, warming within a moderate range, coupled with stronger boundary currents, can expand suitable juvenile and adult habitat and increase year-class strength even for nominally cold-water taxa [66].

Oceanographic factors such as variations in ocean current dynamics may have had a positive effect alongside temperature changes. Notably, the presence of warm-water Mollusks of southern origin has been reported as far north as 43° N about 900–1000 years ago [49], suggesting a temporary strengthening of the northward extension of the Tsushima Current during the late Medieval Warm Period [67]. The strengthened Tsushima Current likely influenced the East Korea Warm Current, increasing both surface and subsurface temperatures in the Southeast Sea compared to adjacent waters [9,68]. Such warming may have increased marine primary productivity in the ocean, increasing the abundance of key prey such as plankton, which could have supported population growth [48].

The more recent historical *N*_e_ shows a sharp decline in *N*_e_ within tens to hundreds of generations. Environmental data were not available in this study; however, the decline of cold-water fish populations along the Korean coast is closely linked to ocean warming [69]. For example, the collapse of the Alaska pollock *Gadus chalcogrammus* stock has been attributed to intensive fishing on juveniles, coinciding with habitat changes driven by rising sea surface temperatures in the East Sea [69,70]. In addition, while catches of cold-water species along the Korean coast have sharply decreased over the past 30 years due to increasing water temperatures, the relative abundance of warm-water species, such as anchovy and squid, has risen markedly [69]. Together, these patterns indicate that ocean climate change is reshaping fish community structure.

Overfishing is also recognized as a major driver of the long-term decline in Korean coastal fish stocks [71]. Korea total fisheries production has been steadily decreasing since the mid-1980s, and FAO and other international organizations have repeatedly warned of global depletion of fishery resources due to overexploitation [69,71]. In line with this, Korea has acknowledged the need to reduce fishing pressure and implement sustainable fisheries management [72].

Population bottlenecks are further known to be affected by recruitment failure and adult mortality [73]. For instance, the recent long-term decline of *Clupea pallasii* has been attributed largely to increased natural mortality in adults [73]. When unfavorable marine conditions impair spawning success and early survival, leading to repeated recruitment failures over several years, populations can collapse rapidly, even when the adult stock initially appears sufficiently abundant [73,74,75]. This decline is likely accelerated by factors such as intensified industrial fishing along the Korean coast, overfishing, habitat disturbance, and rapid ocean thermal anomalies due to climate change [76,77,78,79].

In the short term, *N*_e_ recovery should be prioritized through spawning season catch restrictions, gear and catch management, and protection of juvenile and adult reef habitats. Furthermore, genetic monitoring every 3–5 years should be conducted to verify changes in *N*_e_ and allelic abundance.

## 5. Conclusions

We analyzed seven microsatellite loci in five *Sebastes thompsoni* populations sampled in 2018 and found unexpectedly high genetic diversity, as reflected by observed heterozygosity of 0.759–0.816 and negative *F*_IS_ values that point to persistent gene flow along the Korean coast. Consistent results from STRUCTURE, DAPC, and AMOVA failed to reveal any geographic differentiation, confirming a panmictic population structure. Despite this connectivity, LDNe estimates suggest that current effective population sizes are less than 1000 individuals for all populations, and VarEff analysis suggests a peak expansion approximately 600 years ago and a recent rapid decline, suggesting a risk of diversity loss. These patterns underscore the need to protect habitats and curb overfishing to rebuild and maintain *N*_e_, while stock enhancement programs should rely on broodstocks drawn from genetically diverse sources to safeguard heterozygosity. Collectively, these genetic and demographic indicators provide a robust, evidence-based foundation for managing *S. thompsoni* amid continuing environmental change and fishing pressure.

## Figures and Tables

**Figure 1 biology-14-01559-f001:**
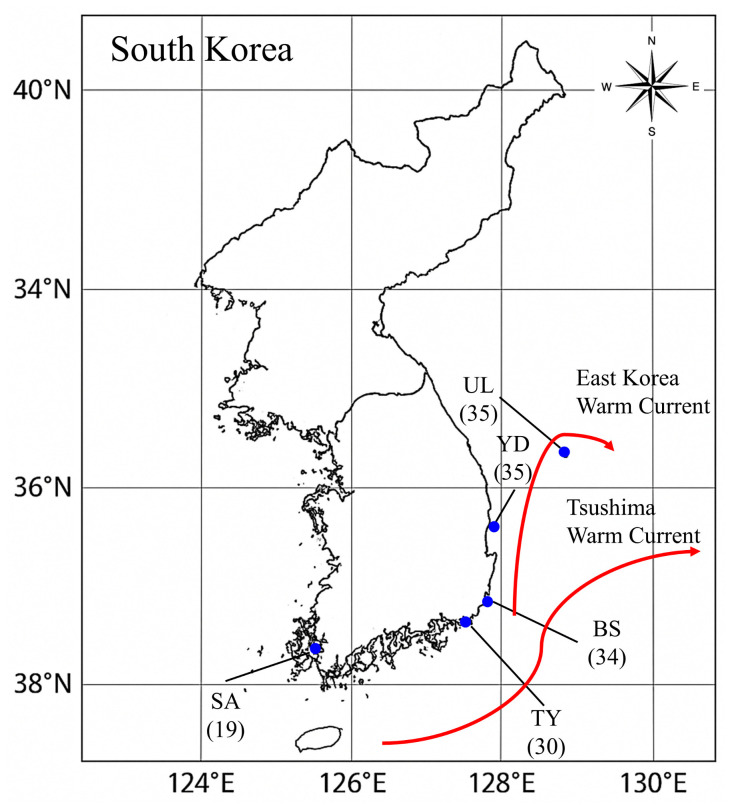
Sampling locations of *S. thompsoni* in the Korean Peninsula. BS, Busan population; SA, Sinan population; TY, Tongyeong; UL, Ulleungdo, YD, Yeongduk. The number in parentheses following the regional abbreviation is the sample size. The red lines represent ocean currents, the Tsushima and East Sea warm currents respectively.

**Figure 2 biology-14-01559-f002:**
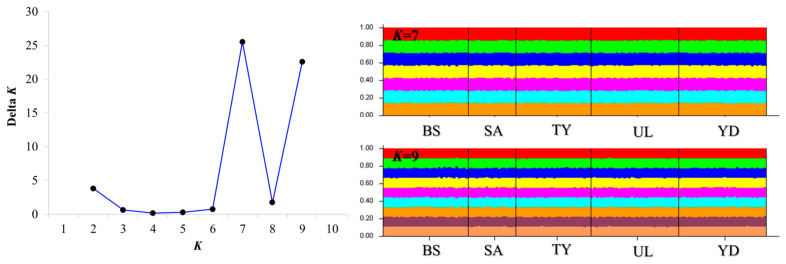
Genetic structure of *S. thompsoni* inferred with STRUCTURE. Left: Evanno’s Δ*K* across *K* = 1–10. Peaks occur at *K* = 7 and *K* = 9. Note that Δ*K* is undefined at *K* = 1. Right: STRUCTURE bar plots for *K* = 7 and 9 under the admixture model (10 replicate runs per *K*; burn-in 10,000; MCMC 100,000). Each vertical bar is an individual; colored segments are membership coefficients (q) for inferred clusters; individuals are grouped by sampling locality (BS, SA, TY, UL, YD). Despite Δ*K* peaks, bar plots show uniform admixture across localities, consistent with panmixia. Together with the log-likelihood profile indicating the best support near *K* = 1 (Table 4), these patterns support a single, genetically single population. Population codes: BS, Busan; SA, Sinan; TY, Tongyeong; UL, Ulleungdo; YD, Yeongduk.

**Figure 3 biology-14-01559-f003:**
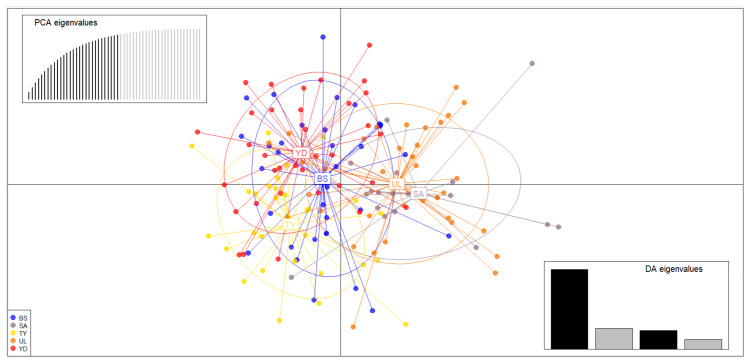
Scatterplots of discriminant analysis of principal components (DAPC). The abbreviated alphabets in the picture are population IDs. Colored dots of different shapes represent individuals from different geographic populations, and the PCA and DA scatterplots on the right side of the graph represent the number of principal components and discriminant functions for calculation. Colored dots represent individual samples in the scatter plot, and clusters of dots indicate a single genetic group. Population codes: BS, Busan; SA, Sinan; TY, Tongyeong; UL, Ulleungdo; YD, Yeongduk.

**Figure 4 biology-14-01559-f004:**
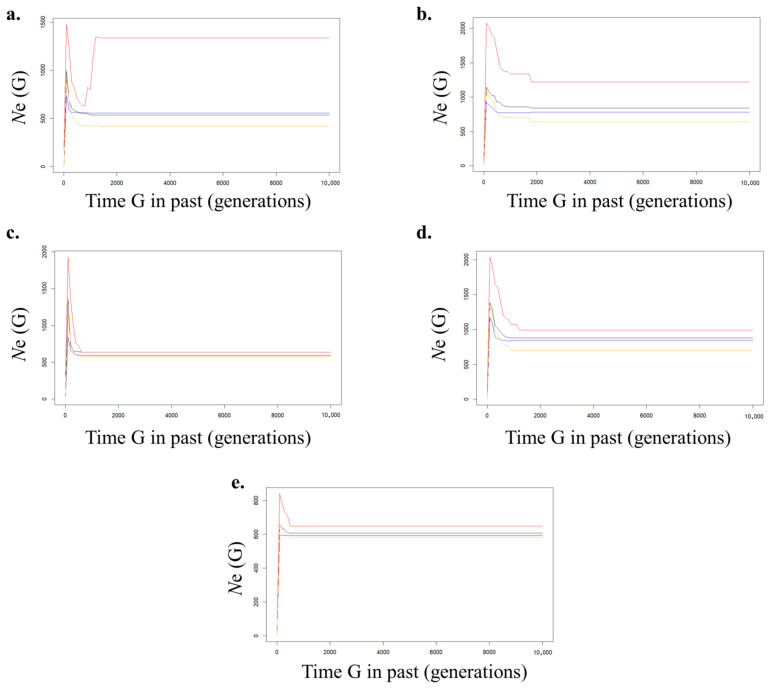
Historical trajectories of effective population size (*N*_e_) inferred by VarEff. Posterior mean (red), mode (blue), median (black) and harmonic mean (orange) of *N*_e_ are shown at each time point. The mean represents the expected value of the posterior distribution, the mode the most probable (highest-density) *N*_e_, the median the 50% quantile, and the harmonic mean the inverse of the average of inverse *N*_e_ values. Time (generations ago) is plotted on the x-axis and *N*_e_ on the y-axis. “Time G” represents the number of generations from the present. G = 0 is the present, and larger values indicate the past. (**a**) BS, Busan population; (**b**) SA, Sinan population; (**c**) TY, Tongyeong; (**d**) UL, Ulleungdo, (**e**) YD, Yeongduk.

**Table 1 biology-14-01559-t001:** Genetic diversity of *S. thompsoni* based on analysis of seven microsatellite loci.

ID	Region Name	*N*	*N* _A_	*A* _R_	*H* _O_	*H* _E_	*P* _HWE_	*F* _IS_
BS	Busan	34	7.3	6.54	0.790	0.680	0.000 ***	−0.160
SA	Sinan	19	6.3	6.29	0.759	0.686	0.000 ***	−0.111
TY	Tongyeong	30	7.0	6.38	0.814	0.699	0.000 ***	−0.168
UL	Ulleungdo	35	7.3	6.55	0.767	0.696	0.000 ***	−0.105
YD	Yeongduk	35	7.3	6.30	0.816	0.659	0.000 ***	−0.243 ***

*N*: number of samples; *N*_A_: average number of alleles; *A*_R_*:* Allelic richness*; H*_O_: observed heterozygosity; *H*_E_: expected heterozygosity; *F*_IS_: Inbreeding coefficient, *P*_HWE_: Hardy-Weinberg equilibrium; *** *p* < 0.001. Population codes: BS, Busan; SA, Sinan; TY, Tongyeong; UL, Ulleungdo; YD, Yeongduk.

**Table 2 biology-14-01559-t002:** Estimates of bottleneck and effective population size for five populations.

Population ID	*N*	Wilcoxon Signed-Rank Test				*N* _e_	(95% CI)
		*P* _IAM_	*P* _TPM_	*P* _SMM_	Mode-Shift		
BS	34	0.008 **	0.039 *	0.078	Shifted	127	(56–∞)
SA	19	0.008 **	0.016 *	0.015 *	Shifted	254	(31–∞)
TY	30	0.016 *	0.016 *	0.023 *	Shifted	-	(69–∞)
UL	35	0.008 **	0.008 **	0.008 *	Shifted	166	(55–∞)
YD	35	0.008 **	0.008 *	0.008 *	Shifted	108	(44–∞)

*N*: number of samples; *N*_e_: effective population size; *P*_IAM_: *p* value of bottleneck test using infinite allele mutation model; *P*_TPM_: *p* value of bottleneck test using two-phase mutation model (10% variance and 90% proportions of SMM); *P*_SMM_: *p* value of bottleneck test using stepwise mutation model; *N*_e_: estimated effective population size using NeEstimator ver. 2.1 software; CI: confidence interval; * *p* < 0.05, ** *p* < 0.01. Population codes: BS, Busan; SA, Sinan; TY, Tongyeong; UL, Ulleungdo; YD, Yeongduk.

**Table 3 biology-14-01559-t003:** Pairwise genetic differentiation of microsatellite (*F*_ST_) values among populations according to microsatellite analysis of *S. thompsoni.* Entries above the diagonal are *p* values for tests of population differentiation; entries below the diagonal are pairwise *F*_ST_. Population codes: BS, Busan; SA, Sinan; TY, Tongyeong; UL, Ulleungdo; YD, Yeongduk.

	BS	SA	TY	UL	YD
BS	-	0.139	0.218	0.322	0.177
SA	0.000	-	0.032	0.995	0.008
TY	0.002	0.008	-	0.163	0.070
UL	0.000	0.000	0.001	-	0.019
YD	0.000	0.008	0.002	0.005	-

Pairwise genetic differentiation of significance level (above); pairwise genetic differentiation of microsatellites (below). Population codes: BS, Busan; SA, Sinan; TY, Tongyeong; UL, Ulleungdo; YD, Yeongduk.

**Table 4 biology-14-01559-t004:** Results of STRUCTURE analysis for estimating the number of genetic clusters (*K*). The table summarizes the estimated log probability of the data [Ln P(D|*K*)], the mean and variance of the likelihood across ten replicate runs, and the mean admixture coefficient (α) for each *K*. A higher (less negative) Ln P(D|K) indicates better model fit, while lower variance reflects more consistent convergence among replicates. Although ΔK values peaked at *K* = 7 and *K* = 9, the log-likelihood pattern and homogeneous admixture across all individuals suggest that the optimal clustering corresponds to *K* = 1, indicating a single, panmictic population.

K	Estimated Ln Prob of Data (L(K))	Mean Value of Ln Likelihood	Variance of Ln Likelihood	Mean Value of Alpha (α)
1	−3331.3	−3316.7	29.1	-
2	−3406.4	−3272.0	268.9	0.9695
3	−3487.6	−3293.1	389.1	1.5947
4	−3383.8	−3302.1	163.4	2.7589
5	−3348.2	−3307.7	80.9	1.697
6	−3352.4	−3307.1	90.6	1.8328
7	−3337.5	−3307.5	60.2	3.844
8	−3333.1	−3310.2	45.8	4.5462
9	−3370.5	−3303.9	133.3	1.5993
10	−3348.7	−3305.1	87.2	2.758

**Table 5 biology-14-01559-t005:** Summary information of the analysis of molecular variance for populations.

Source of Variation	Sum of Squares	Variance Components	Percentage of Variance	*F* _S_ _T_
Among groups	10.311	0.00304	0.13	
Within populations	720.444	2.39350	99.87	0.000
Total	730.755	2.39654	100.00	-

## Data Availability

Microsatellite markers were deposited in Sekino et al. [8] and An et al. [29]. The data sets generated and analyzed during this study are published as Appendix A.

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
