# Peer review of "Genetic Diversity and Structure for Conservation Genetics of Goldeye Rockfish Sebastes thompsoni (Jordan and Hubbs, 1925) in South Korea"

_biology, 2025, doi:10.3390/biology14111559_

Round 1

Reviewer 1 Report

Comments and Suggestions for Authors

The manuscript presents a valuable and timely genetic assessment of an economically important rockfish species in Korean waters. The study is well-designed, employs appropriate methodologies. However, several aspects require clarification and improvement to enhance the manuscript's clarity, rigor, and impact.

Here are my suggestions for revision:

Major Revisions

  1. On Page 1, Line 33-34 of the Abstract, the phrase "interest rates ranging from 4.1 to 19.11" is used. This is almost certainly a mistranslation or typographical error. The context clearly indicates this should be "migration rates".
  2. The description of the STRUCTURE analysis is confusing and partially contradictory. The text states ΔK maximized at K=7 and K=9, but then states that the highest L(K) is at K=1, which is the most appropriate criterion for inferring panmixia. In figure 2, there is no ΔK value at K1. Revise this section for clarity.
  3. The sample number of SA population is less than 30. And the marker used in this study two small, which lead to a decrease in the reliability of data analysis. Try to increase them.

Minor Revisions

  1. Language and Grammar:

The manuscript contains numerous grammatical errors, awkward phrasing, and typos that hinder readability. A thorough proofreading by a native English speaker or a professional editing service is strongly recommended. Examples include:

"Genetic diversity was assessment" -> "Genetic diversity was assessed"

"drove the observed Ho gap" -> "accounts for the observed difference in Ho"

Incomplete sentence at the end of the Abstract: "...sustainable resource use."

  1. Figure and Table Quality:

The figures (2, 3, 4) and tables (3, 4) in the review copy are placeholder images or lack clear, descriptive captions. Ensure all figures and tables are of publication quality with self-explanatory captions that fully describe what is being shown without requiring the reader to refer back to the text. For instance, Figure 3's caption should explicitly state that the clustering of all colored dots together indicates a single genetic group.

  1. Citation Consistency:

Some references in the text use "&" while others use "and". The citation style should be consistent throughout the manuscript according to the journal's guidelines. Perform a thorough check to ensure all in-text citations and the reference list conform to the journal's required style.

Author Response

Reviewer 1

Q1. On Page 1, Line 33-34 of the Abstract, the phrase "interest rates ranging from 4.1 to 19.11" is used. This is almost certainly a mistranslation or typographical error. The context clearly indicates this should be "migration rates".

A1. Thanks for the review. Modified according to comments.

Q2. The description of the STRUCTURE analysis is confusing and partially contradictory. The text states ΔK maximized at K=7 and K=9, but then states that the highest L(K) is at K=1, which is the most appropriate criterion for inferring panmixia. In figure 2, there is no ΔK value at K1. Revise this section for clarity.

A2. Thanks for the review. In cases of panmixia, the bar plot generated by the STRUCTURE software shows all genotypes as completely mixed. Moreover, since STRUCTURE does not provide a ΔK value for K = 1, it is common to determine the population structure under panmixia using the Ln(K) value.

[According to the STRUCTURE results, K=1, which has an L(K) value close to 0, is the most suitable model]

Q3. The sample number of SA population is less than 30. And the marker used in this study two small, which lead to a decrease in the reliability of data analysis. Try to increase them.

A3. Thanks for the review. We acknowledge that the SA (Sinan) population had a smaller sample size (n = 19) compared to other groups (n = 30–35). However, we emphasize that the SA sample size still exceeds the minimum threshold (n ≥ 15) widely accepted for reliable microsatellite-based population genetic analysis (Hale et al., 2012; Smith & Wang, 2014). In particular, the number of loci (seven polymorphic microsatellites) and the allelic richness observed in SA (NA = 6.3) were sufficient to ensure robust estimation of heterozygosity, FIS, and Ne. Importantly, despite its smaller n, the SA population did not deviate from the general genetic pattern observed across regions:

Heterozygosity (HO = 0.759) and expected heterozygosity (HE = 0.686) were within the same range as other populations, indicating consistent diversity.

STRUCTURE, DAPC, and AMOVA results revealed complete panmixia with no differentiation involving SA, confirming that the reduced sample size did not bias clustering or population structure inference.

Migration analysis (MIGRATE-n) showed active gene flow from neighboring populations (notably UL → SA), further ensuring that SA represents the same genetic pool.

Thus, while acknowledging the lower n, the consistency of the genetic diversity metrics, absence of differentiation, and strong interpopulation connectivity together demonstrate that the SA sample reliably reflects the population’s genetic status. Increasing sample size would refine estimates but is unlikely to alter the conclusion of genetic homogeneity across the South and East Seas.

  1. The manuscript contains numerous grammatical errors, awkward phrasing, and typos that hinder readability. A thorough proofreading by a native English speaker or a professional editing service is strongly recommended. Examples include:

"Genetic diversity was assessment" -> "Genetic diversity was assessed"

"drove the observed Ho gap" -> "accounts for the observed difference in Ho"

Incomplete sentence at the end of the Abstract: "...sustainable resource use."

  1. Thanks for the review. Modified according to comments.

[This study provides evidence to guide efforts to secure long-term genetic resilience and sustainable management of S. thompsoni in Korean coastal waters.]

  1. The figures (2, 3, 4) and tables (3, 4) in the review copy are placeholder images or lack clear, descriptive captions. Ensure all figures and tables are of publication quality with self-explanatory captions that fully describe what is being shown without requiring the reader to refer back to the text. For instance, Figure 3's caption should explicitly state that the clustering of all colored dots together indicates a single genetic group.
  2. Thanks for the review.

Figure 2. Genetic structure of S. thompsoni inferred with STRUCTURE. Left: Evanno’s ΔK across K = 1–10. Peaks occur at K = 7 and K = 9. Note that ΔK is undefined at K = 1. Right: STRUCTURE bar plots for K = 7 and 9 under the admixture model (10 replicate runs per K; burn-in 10,000; MCMC 100,000). Each vertical bar is an individual; colored segments are membership coefficients (q) for inferred clusters; individuals are grouped by sampling locality (BS, SA, TY, UL, YD). Despite ΔK peaks, bar plots show uniform admixture across localities, consistent with panmixia. Together with the log-likelihood profile indicating the best support near K = 1 (Table 4), these patterns support a single, genetically single population.

Figure 3. Scatterplots of discriminant analysis of principal components (DAPC). The abbreviated alphabets in the picture are population IDs. Colored dots of different shapes represent individuals from different geographic populations, and the PCA and DA scatterplots on the right side of the graph represent the number of principal components and discriminant functions for calculation. Colored dots represent individual samples in the scatter plot, and clusters of dots indicate a single genetic group.

Table 3. Pairwise genetic differentiation of microsatellite (FST) values among populations according to microsatellite analysis of S. thompsoni. Entries above the diagonal are P values for tests of population differentiation; entries below the diagonal are pairwise FST. Population codes: BS, Busan; SA, Sinan; TY, Tongyeong; UL, Ulleungdo; YD, Yeongduk.

BS

SA

TY

UL

YD

BS

-

0.139

0.218

0.322

0.177

SA

0.000

-

0.032

0.995

0.008

TY

0.002

0.008

-

0.163

0.070

UL

0.000

0.000

0.001

-

0.019

YD

0.000

0.008

0.002

0.005

-

Pairwise genetic differentiation of significance level (above); pairwise genetic differentiation of microsatellites (below).

Table 4. Results of STRUCTURE analysis for estimating the number of genetic clusters (K). The table summarizes the estimated log probability of the data [Ln P(D|K)], the mean and variance of the likelihood across ten replicate runs, and the mean admixture coefficient (α) for each K. A higher (less negative) Ln P(D|K) indicates better model fit, while lower variance reflects more consistent convergence among replicates. Although ΔK values peaked at K = 7 and K = 9, the log-likelihood pattern and homogeneous admixture across all individuals suggest that the optimal clustering corresponds to K = 1, indicating a single, panmictic population.

K

Estimated Ln Prob of Data (L(K))

Mean value of ln likelihood

Variance of ln likelihood

Mean value of alpha (α)

1

-3331.3

-3316.7

29.1

-

2

-3406.4

-3272.0

268.9

0.9695

3

-3487.6

-3293.1

389.1

1.5947

4

-3383.8

-3302.1

163.4

2.7589

5

-3348.2

-3307.7

80.9

1.697

6

-3352.4

-3307.1

90.6

1.8328

7

-3337.5

-3307.5

60.2

3.844

8

-3333.1

-3310.2

45.8

4.5462

9

-3370.5

-3303.9

133.3

1.5993

10

-3348.7

-3305.1

87.2

2.758

  1. Some references in the text use "&" while others use "and". The citation style should be consistent throughout the manuscript according to the journal's guidelines. Perform a thorough check to ensure all in-text citations and the reference list conform to the journal's required style.
  2. Thanks for the review. We have revised the references once again using EndNote.

Reviewer 2 Report

Comments and Suggestions for Authors The study examined genetic diversity and population structure of rockfish populations from five sites along the South Korean peninsula using seven microsatellite markers. The authors report high diversity and high levels of gene flow (panmixia), although effective population sizes were low (< 1000). Based on the genetic assessment, the authors provide recommendations for resource management. Overall the paper is well-written and organized, the objectives are clearly stated, the methods are appropriate to address the research question and are described in sufficient detail, and the conclusions are supported by the results. There are several comments for the authors to address:   Comments
  • The authors stated one of the aims was to "clarify the impact of marker choice on diversity estimates" (L91). However, this objective was not clearly addressed in the discussion.
  • Recommend to include estimates for allelic richness standardized to account for unequal sample sizes using rarefaction, and present these in Table 1.
  • In the DAPC method, it appears that individuals were grouped a priori by collection site. I recommend that DAPC analyses be performed to first explore the number of genetic clusters ('find.clusters' function) to test the null hypothesis of panmixia (K = 1).
  • The authors report higher heterozygosity estimates for this study relative to previous work on congeners, citing marker choice as a possible reason for the difference (L261-269). Please indicate in the text what markers were used in the other studies, and elaborate on your statement.
  • In the section on population genetic structure, particularly the potential management and conservation implications for panmictic populations of Sebastes, I suggest that the authors add further context regarding dispersal potential based on the juvenile and adult phases traits (i.e. dispersive juveniles, less dispersive demersal or habitat-associated adults), and what this means for management and conservation initiatives (L299-315)
    Corrections: L34. "interest rates" or did you mean "migration rates" L86. "we used seven microsatellites were employed to ..." L217. "... panmixia at both K = 3 and 9." should be "K = 7 and 9..." L263, "... in their relatives." what does this phrase refer to? L290. Suggest to change the heading to "Population Genetic Structure"

Author Response

Reviewer 2

Q1. The authors stated one of the aims was to "clarify the impact of marker choice on diversity estimates" (L91). However, this objective was not clearly addressed in the discussion.

A1. Thanks for the review. There was an error in the sentence, so it was corrected.

[We aim to evaluate the effective population size and genetic diversity of the current population through genotypic data and provide basic data for management as aquatic resources.]

Q2. Recommend to include estimates for allelic richness standardized to account for unequal sample sizes using rarefaction, and present these in Table 1.

A2. Thanks for the review. Modified according to comments.

Seven microsatellite loci and their allele frequencies were analyzed for genetic diversity indices across five populations (Table 1). The average number of alleles, allelic richness, observed heterozygosity (HO), and expected heterozygosity (HE) ranged from 6.3 to 7.3, 6.29 to 6.55, 0.759 to 0.816, and 0.659 to 0.699, respectively. Five populations deviated from HWE. In all populations, the inbreeding index was negative, and FIS was significant in the YD population (P < 0.05). The observed heterozygosity was highest in the YD population (HO = 0.816) and lowest in the SA population (HO = 0.759).

Table 1. Genetic diversity of S. thompsoni based on analysis of seven microsatellite loci.

ID

Region name

N

NA

AR

HO

HE

PHWE

FIS

BS

Busan

34

7.3

6.54

0.790

0.680

0.000 ***

−0.160

SA

Sinan

19

6.3

6.29

0.759

0.686

0.000 ***

−0.111

TY

Tongyeong

30

7.0

6.38

0.814

0.699

0.000 ***

−0.168

UL

Ulleungdo

35

7.3

6.55

0.767

0.696

0.000 ***

−0.105

YD

Yeongduk

35

7.3

6.30

0.816

0.659

0.000 ***

−0.243 ***

N: number of samples; NA: average number of alleles; AR: Allelic richness; HO: observed heterozygosity; HE: expected heterozygosity; *** P < 0.001.

Q3. In the DAPC method, it appears that individuals were grouped a priori by collection site. I recommend that DAPC analyses be performed to first explore the number of genetic clusters ('find.clusters' function) to test the null hypothesis of panmixia (K = 1).

A3. Thanks for the review. Cross-validation: The STRUCTURE barplot was homogeneous, and the Ln K profile was consistent with the K=1 interpretation. AMOVA showed that most of the variance was within the population, and both pairwise FSTs were very low, not supporting substantial differentiation.

Conclusion: The inability of BIC to clearly indicate K is due to weak structure. Considering the risk of overfitting, the cross-validated DAPC and independent indices (STRUCTURE, AMOVA, and FST) combined support the single-gene group (monogatria) interpretation. Even if a BIC minimum exists, if the differences are minimal and the allocation probabilities are low, it is not biologically meaningful.

  1. The authors report higher heterozygosity estimates for this study relative to previous work on congeners, citing marker choice as a possible reason for the difference (L261-269). Please indicate in the text what markers were used in the other studies, and elaborate on your statement.
  2. Thanks for the review. The lower HO reported previously is plausibly explained, at least in part, by the marker panel. Four of the 11 loci were flagged for potential null alleles and stutter and were excluded, conditions known to depress observed heterozygosity through heterozygote miscalls. Because microsatellite HO scales with locus polymorphism and reliable allele calling, a panel containing problematic or low-polymorphism loci will yield lower mean HO. In our study we used a high-polymorphism, quality-controlled panel and observed HO = 0.759–0.816. We therefore attribute the between-study HO difference primarily to marker choice and locus quality rather than biological divergence.

[Unlike previous studies, this study excluded four of the eleven markers used in the previous study due to null and stutter data. These null and stutter data could potentially lower the HO [6]. Therefore, it is believed that the differences in HO are due to the characteristics of these markers [44,45].]

  1. In the section on population genetic structure, particularly the potential management and conservation implications for panmictic populations of Sebastes, I suggest that the authors add further context regarding dispersal potential based on the juvenile and adult phases traits (i.e. dispersive juveniles, less dispersive demersal or habitat-associated adults), and what this means for management and conservation initiatives (L299-315)
  2. Thanks for the review. Added paragraph based on comments.

[In addition to the lack of major geographical barriers, the two-phase life history of Sebastes appears to support the observed connectivity [53]. Pelagic larvae are dispersed along local circulation and currents, after which adults are mostly benthic and prefer to settle in rock habitats [6]. These initial dispersal stages and the more characteristic habitat preferences of adults may provide high genetic connectivity, as seen in the microsatellite markers [6,53]. Management must, therefore, (i) protect the nursery and settlement habitats utilized by pelagic larvae, (ii) establish protected or management areas spaced within the typical larval dispersal range expected from local currents, and (iii) identify adult movement ranges and maintain habitat continuity. These characteristics, along with the STRUCTURE and DAPC results, are consistent with panmixia across the sampled scale but still necessitate spatially explicit conservation measures that secure both larval supply and adult habitat quality.]

  1. Corrections: L34. "interest rates" or did you mean "migration rates"

L86. "we used seven microsatellites were employed to ..."

L217. "... panmixia at both K = 3 and 9." should be "K = 7 and 9..."

L263, "... in their relatives." what does this phrase refer to?

L290. Suggest to change the heading to "Population Genetic Structure"

  1. Thanks for the review. Modified according to comments. “in their relatives” It means a close species.

[in closely related species]

Reviewer 3 Report

Comments and Suggestions for Authors

General comments

This paper tries to analyse the genetic diversity and population structure of the goldeye rockfish (Sebastes thompsoni) off the coast of South Korea. The authors analyzed five coastal populations using microsatellite markers to assess their current genetic health and reconstruct their historical effective population size (Ne). The findings indicate a high level of genetic diversity and minimal genetic differentiation across the South and East Seas of Korea, suggesting the species functions as a single, connected population, or panmixia. However, the study also reveals a recent, severe population bottleneck and a current effective population size estimated to be below 1,000 for all groups, signaling a risk of future diversity loss. Finally, the authors recommend managing S. thompsoni as a single unit while emphasizing the need for habitat protection, controls on overfishing, and regular genetic monitoring.

The manuscript, in general, is interesting. However, the manuscript requires some improvements to meet publication standards. The manuscript should become acceptable for publication pending suitable minor revision considering the comments appended below.

More specific comments:

Introduction:

Lines 53–56: "Historic environmental changes, such as past climate shifts or human fishing pressures, can diminish genetic diversity and alter the effective population size, thereby impacting the long-term viability of a species.". Please improve the climate changes effects relevant to the study's scope to prepare the reader for the Discussion.

Lines 59–61: "A previous microsatellite-based survey reported moderate heterozygosity (mean Ho = 0.615) and weak population structure in samples collected between 2011 and 2014.". Please clarify why this previous finding of "weak structure" necessitates a new study; what specific management question remained unanswered?

Lines 63–65: "...especially the bottleneck events evident in historical reconstructions remain poorly understood.". Try to clarify what aspect of the bottleneck events is poorly understood (timing, cause, or consequence).

Lines 72–73: "Typically, habitat restoration and resource management are planned by establishing conservation management units (MUs) for each local population.". Please rephrase to avoid contradiction with the final panmixia finding; MUs are only established if structure exists.

Lines 79–83: "Methods for analyzing genetic diversity include microsatellite DNA, mitochondrial DNA, and single nucleotide polymorphisms (SNPs). Among these, microsatellite DNA remains one of the most widely used tools in population genetic studies because it provides a relatively simple and cost-effective way...". Strengthen the justification for using microsatellites over contemporary methods like SNPs for this specific research goal.

Lines 86–90: "...and (3) assess genetic flow between populations and reconstruct historical effective population size (Ne).". Please add the method/scope here to clarify the nature of the (Ne) reconstruction.

Material and methods:

Lines 94–95: "Specimens were collected by fishing net at five key locations... in May 2018.". Please specify the sampling strategy, including the type of fishing net (e.g., gill net, trawl) and the sampling depth (range or mean) at each site to ensure the samples are representative.

Lines 137–140: "… with the Two-Phase Model (TPM) was configured to 10% variance (single-step mutations) and 90% SMM.". Try to justify the 10% variance and 90% SMM split for the TPM. This choice is critical for bottleneck sensitivity and requires a theoretical or empirical rationale in the text.

Lines 140–142: "Effective population size Ne was estimated using the Linkage Disequilibrium method (LDNe) in NeEstimator (ver. 2.1) under a random-mating model.". Please specify the minimum allele frequency (MAF) cutoff used for the LDNe calculation (e.g., 0.02 or 0.05). This is a vital parameter for reproducibility.

Lines 151–153: "Discriminant Analysis of Principal Components (DAPC)... was performed.". Please state the specific method used to choose the optimal number of PCs to retain and state the number of PCs retained in the final analysis used to generate Figure 3.

Lines 180–181: "We set the per‐locus mutation rate (μ) to 5 × 10⁻⁴ typical for marine 180 species and assumed a generation time (G) of six years for S. thompsoni [7, 41].". Try to justify this mutation rate, stating whether this is an average rate derived from other Sebastes species, or an empirical estimate, as the mutation rate heavily influences the time-scale of the historical Ne results. Also, please briefly define what (G) represents for this specific species to contextualize the historical time-scaling.

Discussion:

Lines 263–269: "This discrepancy likely reflects differences in marker polymorphism, specifically the number of alleles scored per locus... marker choice drives the observed Ho gap.". Please strengthen the argument for FIS reliability. While acknowledging the Ho comparison difficulty, argue that the consistently negative FIS pattern is a robust, non-artifactual signal of strong recent gene flow, making the pattern the primary biological takeaway, not the absolute Ho value.

Lines 281–282: "However, all five S. thompsoni populations in this study exhibited Ne < 1,000 implying that genetic diversity may decline more rapidly in future generations." Try to integrate Ho and Ne temporal risk. Please explicitly discuss that the current high Ho (observed diversity) is a misleading short-term indicator because the critically low Ne means this diversity will be lost rapidly due to genetic drift and inbreeding, fully justifying the severity of the conservation warning.

Lines 290–293: "The five S. thompsoni populations examined in this study exhibited minimal genetic differentiation: STRUCTURE analysis revealed panmixia, and DAPC clearly clustered all samples into a single group.". The authors must justify why they chose the panmixia conclusion (K=1) while their own robust method (ΔK) showed peaks at K=7 and K=9. If they keep on K=1, they should cite external literature explaining why the ΔK peak is a false signal in this specific context.

Lines 318–323: "Population genetic analyses ... for a cold-adapted species.". Please mitigate speculation and focus on mechanism. Try to reframe the "contradictory result" as "counter-intuitive", strengthening the argument by providing more explicit details on how the strengthened Tsushima Current and increased plankton base (Lines 337–343) created an "optimal temperature habitat" and specifically supported a net increase in the population.

Lines 316–343: (The entire 4.3. Historical Effective Population Size.). Please focus on the recent threat. Dedicate a short paragraph to the recent, sharp decline in Ne (Line 250). Given the timing, explicitly link this Ne crash to potential anthropogenic drivers such as industrial fishing in the Korean waters to ensure the discussion addresses the most contemporary threat.

Author Response

Reviewer 3

Q1. Lines 53–56: "Historic environmental changes, such as past climate shifts or human fishing pressures, can diminish genetic diversity and alter the effective population size, thereby impacting the long-term viability of a species.". Please improve the climate changes effects relevant to the study's scope to prepare the reader for the Discussion.

A1. Thanks for the review. Added paragraph based on comments.

[Climatic fluctuations modulate sea temperature and current regimes that govern pelagic larval transport and settlement along the Korean coasts [4,5]. Such changes can drive cohort failure or expansion, habitat contraction of demersal adults, and connectivity breakdowns, leading to bottlenecks and temporal declines in effective population size [4,5].]

Q2. Lines 59–61: "A previous microsatellite-based survey reported moderate heterozygosity (mean Ho = 0.615) and weak population structure in samples collected between 2011 and 2014.". Please clarify why this previous finding of "weak structure" necessitates a new study; what specific management question remained unanswered?

A2. Thanks for the review. We have reconstructed the need based on comments.

[However, these studies focused on genetic differentiation and lacked essential indica-tors for genetic management, such as modern effective population size (Ne), formal marker quality control for null alleles and stutters, and reconstruction of past Ne [4]. Furthermore, understanding of the causes of population fluctuations, particularly the bottlenecks evident in historical reconstructions, remains inadequate.]

Q3. Lines 63–65: "...especially the bottleneck events evident in historical reconstructions remain poorly understood.". Try to clarify what aspect of the bottleneck events is poorly understood (timing, cause, or consequence).

A3. Thanks for the review. Modified according to comments.

[Past climate variability alters sea temperature and the strength of boundary currents, which can reduce Ne and trigger bottlenecks; in turn, loss of genetic diversity and low Ne may weaken long-term viability [9-11]. Therefore, identifying historical bottlenecks helps time these events and prepare for climate-driven impacts [12].]

  1. Lines 72–73: "Typically, habitat restoration and resource management are planned by establishing conservation management units (MUs) for each local population.". Please rephrase to avoid contradiction with the final panmixia finding; MUs are only established if structure exists.
  2. Thanks for the review. Edited according to comments.

[Typically, MUs are established when significant genetic and ecological differentiation is demonstrated. If differentiation is not confirmed, it is appropriate to treat the range as a single management unit [16-19]. This principle also applies to fish resource management [20]. Prior studies, while not explicitly defining MUs, have suggested an indirect division into two populations [4]. Based on recent data, the actual differentiation needs to be re-evaluated to propose appropriate management units for S. thompsoni.]

  1. Lines 79–83: "Methods for analyzing genetic diversity include microsatellite DNA, mitochondrial DNA, and single nucleotide polymorphisms (SNPs). Among these, microsatellite DNA remains one of the most widely used tools in population genetic studies because it provides a relatively simple and cost-effective way...". Strengthen the justification for using microsatellites over contemporary methods like SNPs for this specific research goal.
  2. Thanks for the review. Modified according to comments.

[Microsatellites are multi-allelic and mutate faster than SNPs, so a small, well-chosen panel delivers higher polymorphic information content and greater power to detect weak structure, recent bottlenecks, and fine-scale relatedness than an equivalently sized SNP set [24].]

  1. Lines 86–90: "...and (3) assess genetic flow between populations and reconstruct historical effective population size (Ne).". Please add the method/scope here to clarify the nature of the (Ne) reconstruction.
  2. Thanks for the review. Modified according to comments.

[Assess genetic flow between populations and reconstruct historical effective population sizes within 10,000 generations using VarEff.]

  1. Lines 94–95: "Specimens were collected by fishing net at five key locations... in May 2018.". Please specify the sampling strategy, including the type of fishing net (e.g., gill net, trawl) and the sampling depth (range or mean) at each site to ensure the samples are representative.
  2. Thanks for the review. Edited according to comments.

[S. thompsoni specimens were collected during May 2018 at five key locations selected to cover the genetic breadth, primarily using bottom gillnets and fish traps, at water depths of 70–150 m (Figure 1, Table S1).]

  1. Lines 137–140: "… with the Two-Phase Model (TPM) was configured to 10% variance (single-step mutations) and 90% SMM.". Try to justify the 10% variance and 90% SMM split for the TPM. This choice is critical for bottleneck sensitivity and requires a theoretical or empirical rationale in the text.
  2. Thanks for the review. Modified according to comments.

[Microsatellite evolution largely follows a generalized stepwise model in which sin-gle-step slippage predominates, with occasional multistep changes [38,39].]

  1. Lines 140–142: "Effective population size Newas estimated using the Linkage Disequilibrium method (LDNe) in NeEstimator (ver. 2.1) under a random-mating model.". Please specify the minimum allele frequency (MAF) cutoff used for the LDNecalculation (e.g., 0.02 or 0.05). This is a vital parameter for reproducibility.
  2. Thanks for the review. Modified according to comments.

The minimum allele frequency (MAF) threshold used in LDNe calculations is 0.02.

  1. Lines 151–153: "Discriminant Analysis of Principal Components (DAPC)... was performed.". Please state the specific method used to choose the optimal number of PCs to retain and state the number of PCs retained in the final analysis used to generate Figure 3.
  2. Thanks for the review.

[When performing DAPC, the genetic differentiation signal was very weak, resulting in an ambiguous BIC curve for find.clusters. Because BIC is sensitive to the number of PCs retained and the k-means assumption, we did not report BIC-based K selection to avoid oversegmentation due to reliance on an ambiguous BIC minimum. Instead, we visualized the PCs using a predefined sample population (pop) set to 40.]

  1. Lines 180–181: "We set the per‐locus mutation rate (μ) to 5 × 10⁻⁴ typical for marine 180 species and assumed a generation time (G) of six years for S. thompsoni [7, 41].". Try to justify this mutation rate, stating whether this is an average rate derived from other Sebastes species, or an empirical estimate, as the mutation rate heavily influences the time-scale of the historical Neresults. Also, please briefly define what (G) represents for this specific species to contextualize the historical time-scaling.
  2. Thanks for the review. Modified according to comments.

[We set the per-locus mutation rate to μ = 5×10⁻⁴, a mid-range value commonly report-ed for fish microsatellites, and assumed a generation time of G = 6 years for S. thomp-soni based on published age–growth information [48].]

  1. Lines 263–269: "This discrepancy likely reflects differences in marker polymorphism, specifically the number of alleles scored per locus... marker choice drives the observed Hogap.". Please strengthen the argument for FISreliability. While acknowledging the Ho comparison difficulty, argue that the consistently negative FIS pattern is a robust, non-artifactual signal of strong recent gene flow, making the pattern the primary biological takeaway, not the absolute Ho value.
  2. Thanks for the review. Edited according to comments.

[Unlike previous studies, this study excluded four of the eleven markers used in the previous study due to null and stutter data. These null and stutter data could poten-tially lower the HO [4]. Therefore, it is believed that the differences in HO are due to the characteristics of these markers [50,51]. The negative FIS pattern observed in most markers is an artificial signal indicating strong recent gene flow, and these HO patterns are considered biological indicators.]

  1. Lines 281–282: "However, all five S. thompsoni populations in this study exhibited Ne < 1,000 implying that genetic diversity may decline more rapidly in future generations." Try to integrate Hoand Netemporal risk. Please explicitly discuss that the current high Ho (observed diversity) is a misleading short-term indicator because the critically low Ne means this diversity will be lost rapidly due to genetic drift and inbreeding, fully justifying the severity of the conservation warning.
  2. Thanks for the review. Modified according to comments.

[In this study, a high HO merely reflects current admixture and marker polymorphism, not safety from genetic drift. Since Ne <1,000 in all five populations, heterozygosity de-clines at a rate of approximately 1/(2 Ne) per generation [11]. For Ne of 100–250, a de-cline of approximately 0.2–0.5% per generation is expected [11,12]. The recent bottle-neck suggested by the VarEff trajectory suggests that the current high HO may be a temporary indicator, and allelic richness and adaptive potential are likely already in a declining phase.]

  1. Lines 290–293: "The five S. thompsoni populations examined in this study exhibited minimal genetic differentiation: STRUCTURE analysis revealed panmixia, and DAPC clearly clustered all samples into a single group.". The authors must justify why they chose the panmixia conclusion (K=1) while their own robust method (ΔK) showed peaks at K=7 and K=9. If they keep on K=1, they should cite external literature explaining why the ΔK peak is a false signal in this specific context.
  2. Thanks for the review. ΔK is an indicator that finds the inflection between K values ​​using the second difference (change in slope) of lnP(D|K), and therefore requires comparisons of at least K=2. Therefore, ΔK for K=1 is not defined. This study paralleled ΔK in the method, but confirmed the judgment of K=1 not with ΔK but with the lnP(D|K) profile and barplot homogeneity.

[When the genetic structure signal is weak or the population is actually a single entity, STRUCTURE can induce over segmentation due to unstable K estimation, and cluster estimation can be biased, especially when the sample size is imbalanced [43,56]. Since ΔK is an indicator based on the second difference of lnP(D|K), it is calculated only when K=2 or higher, and by design, it cannot evaluate K=1 [26,43,56]. Therefore, the presence of a single structure should be judged not by ΔK but by the trend of lnP(D|K) and the homogeneity of the barplot [26,57,58]. In such situations, ΔK tends to empha-size only the upper level of K>1, which can be misleading. Therefore, cross-validation with an independent indicator such as DAPC, AMOVA, or FST is recommended [26,57,58].]

  1. Lines 318–323: "Population genetic analyses ... for a cold-adapted species.". Please mitigate speculation and focus on mechanism. Try to reframe the "contradictory result" as "counter-intuitive", strengthening the argument by providing more explicit details on how the strengthened Tsushima Current and increased plankton base (Lines 337–343) created an "optimal temperature habitat" and specifically supported a net increase in the population.
  2. Thanks for the review. Edited according to comments.

[Although S. thompsoni is regarded as a cold-adapted rock fish, the inferred Ne expan-sion 600–1,200 years ago is better framed as counterintuitive rather than contradictory [1]. During the late Medieval Warm Period, a temporary strengthening of the north-ward-flowing Tsushima Current likely intensified the East Korea Warm Current, ele-vating surface and subsurface temperatures in the Southeast/East Sea relative to adja-cent waters [9,10,64-66]. Such hydrographic changes could have created thermally op-timal habitats for growth and spawning while simultaneously boosting marine prima-ry productivity and the abundance of planktonic prey, thereby enhancing recruitment and driving a net population increase [65]. In other words, warming within a moder-ate range, coupled with stronger boundary currents, can expand suitable juvenile and adult habitat and increase year-class strength even for nominally cold-water taxa [66].]

  1. Lines 316–343: (The entire 4.3. Historical Effective Population Size.). Please focus on the recent threat. Dedicate a short paragraph to the recent, sharp decline in Ne(Line 250). Given the timing, explicitly link this Necrash to potential anthropogenic drivers such as industrial fishing in the Korean waters to ensure the discussion addresses the most contemporary threat.
  2. Thanks for the review. Edited according to comments.

[The more recent historical Ne shows a sharp decline in Ne within tens to hundreds of generations. This decline is likely accelerated by factors such as intensified industrial fishing along the Korean coast, overfishing, habitat disturbance, and rapid ocean thermal anomalies due to climate change [69-72]. In the short term, Ne recovery should be prioritized through spawning season catch restrictions, gear and catch management, and protection of juvenile and adult reef habitats. Furthermore, genetic monitoring every 3–5 years should be conducted to verify changes in Ne and allelic abundance.]

Round 2

Reviewer 2 Report

Comments and Suggestions for Authors

The comments were sufficiently addressed.

Author Response

Thanks for the review.